# Latent Diffusion Planning for Imitation Learning

Amber Xie [1]   Oleh Rybkin [2]   Dorsa Sadigh [1]   Chelsea Finn [1]

## Abstract

Recent progress in imitation learning has been enabled by policy architectures that scale to complex visuomotor tasks, multimodal distributions, and large datasets. However, these methods often rely on learning from large amount of expert demonstrations. To address these shortcomings, we propose Latent Diffusion Planning (LDP), a modular approach consisting of a planner which can leverage *action-free* demonstrations, and an inverse dynamics model which can leverage *suboptimal* data, that both operate over a learned latent space. First, we learn a compact latent space through a variational autoencoder, enabling effective forecasting of future states in image-based domains. Then, we train a planner and an inverse dynamics model with diffusion objectives. By separating planning from action prediction, LDP can benefit from the denser supervision signals of suboptimal and action-free data. On simulated visual robotic manipulation tasks, LDP outperforms state-of-the-art imitation learning approaches, as they cannot leverage such additional data. [1]

## 1. Introduction

Combining large-scale expert datasets and powerful imitation learning policies has been a promising direction for robot learning. Recent methods using transformer backbones or diffusion heads (Octo Model Team et al., 2024; Kim et al., 2024; Zhao et al., 2024; Chi et al., 2023) have capitalized on new robotics datasets pooled together from many institutions (Khazatsky et al., 2024; Open X-Embodiment Collaboration et al., 2023), showing potential for learning generalizable robot policies. However, this recipe is fundamentally limited by expert data, as robotics demonstration data can be challenging, time-consuming, and expensive to

collect. While it is often easier to collect in-domain data that is suboptimal or action-free, these methods are not designed to use such data, as they rely on directly modeling optimal actions.

Prior works in offline RL, reward-conditioned policies, or imitation learning from suboptimal demonstrations attempt to leverage suboptimal trajectories, though they are still unable to utilize action-free data. Notably, these works often make restrictive assumptions like access to either reward labels (Kumar et al., 2020; Chen et al., 2021; Kumar et al., 2019a), the optimality of demonstrations (Beliaev et al., 2022), or similar metrics (Zhang et al., 2022), which can be impractical or noisy to label. Other works implicitly attempt to use unlabelled, suboptimal data via pretraining on such data and later fine-tuning the policy on the optimal data (Radosavovic et al., 2023; Wu et al., 2023b; Cui et al., 2024). While these approaches can potentially learn representations during pretraining, it does not necessarily improve planning capabilities of these methods.

Our key idea is to take a modular approach, where we separate learning a video planner from learning an inverse dynamics model. Each one of these two components can leverage different types of data. For instance, a planner can benefit from action-free data, while an IDM can leverage unlabelled suboptimal data. While using a modular approach has been proposed in recent prior work (Du et al., 2023a; Black et al., 2023), prior approaches focus on high-level decision making by forecasting subgoals, limiting its capabilities for closed loop re-planning in robotics tasks. These works also operate across images, which are high-dimensional and expensive to generate. To create an efficient modular approach that can benefit from all forms of data (suboptimal, action-free, and optimal), we propose learning the planner and inverse-dynamics model over a learned, compact latent space, allowing for closed-loop robot policies. Our imitation learning objective consists of forecasting a dense trajectory of latent states, scaling up gracefully to vision-based domains without the computational complexities of video generation.

We propose Latent Diffusion Planning (LDP), which learns a planner that can be trained on action-free data; and an inverse dynamics model (IDM) that can be trained on data that may be suboptimal. First, it trains a variational autoencoder

---

[1]Stanford [2]UC Berkeley. Correspondence to: Amber Xie <amberxie@stanford.edu>.

*Proceedings of the $42^{nd}$ International Conference on Machine Learning*, Vancouver, Canada. PMLR 267, 2025. Copyright 2025 by the author(s).

[1]Project Website and Code: https://amberxie88.github.io/ldp/

**LDP can use diverse data sources for enhanced training**

**Latent Diffusion Planning**

*Figure 1.* Latent Diffusion Planning. *Left*: LDP separates the control problem into forecasting future states with a diffusion-based planner, and extracting actions with a diffusion-based inverse dynamics model (IDM). This design enables training on heterogeneous sources of data, including suboptimal data and action-free data. *Right*: Unlike action imitation methods such as diffusion policy, LDP is based on forecasting a dense temporal sequence of latent states as well as actions. Using powerful diffusion models for both of these objectives enables LDP to have competitive performance to state-of-the-art imitation learning. Further, unlike prior work on forecasting subgoals, LDP predicts a dense temporal sequence of latent states, which enables scalable closed-loop planning.

with an image reconstruction loss, producing compressed latent embeddings that are used by the planner and inverse dynamics model. Then, it learns an imitation learning policy through two components: (1) a planner, which consumes demonstration state sequences, which may be action-free, and (2) an inverse dynamics model, trained on in-domain, possibly suboptimal, environment interactions. As diffusion objectives have proven to be effective for imitation learning in robotics tasks (Chi et al., 2023), we use diffusion for both forecasting plans (planner) and extracting actions (IDM), which enables competitive performance. Our method is closed-loop and reactive, as planning over latent space is much faster than generating visually and physically consistent video frames.

In summary, our main contributions are threefold:

- We propose a novel imitation learning algorithm, Latent Diffusion Planning, a simple, diffusion planning-based method comprised of a learned visual encoder, latent planner, and an inverse dynamics model.

- We show that Latent Diffusion Planning can be trained on suboptimal or action-free data, and improves from learning on such data in the regime where demonstration data is limited.

- We experimentally show that our method outperforms prior video planning-based work by leveraging temporally dense predictions in a latent space, which enables fast inference for closed-loop planning.

## 2. Related Work

**Imitation Learning in Robotics.** One common approach to learning robot control policies is imitation learning, where policies are learned from expert-collected demonstration datasets. This is most commonly done via behavior cloning, which reduces policy learning to a supervised learning objective of mapping states to actions. Recently, Diffusion Policy (Chi et al., 2023) and Action Chunking with Transformers (Zhao et al., 2023) have shown successful results in complex manipulation tasks using action chunking and more expressive architectures. Diffusion models have also been successful in capturing multimodal human behavior (Pearce et al., 2023). Similarly, Behavior Transformer (Shafiullah et al., 2022) and VQ-BeT (Lee et al., 2024) improve the ability of policies to capture multimodal behaviors. In this work, we focus on forecasting a sequence of future states instead of actions, and use diffusion to capture multimodal trajectories.

**Learning from Unlabelled Suboptimal and Action-Free Data.** Learning from suboptimal data has long been a goal of many robot learning methods, including reinforcement learning. A typical approach is offline reinforcement learning, which considers solving a Markov decision process from an offline dataset of states, actions, and reward (Levine et al., 2020; Kumar et al., 2020; Kostrikov et al., 2021; Hansen-Estruch et al., 2023; Yu et al., 2022). Particularly relevant are the approaches that use supervised learning conditioned on rewards (Schmidhuber, 2019; Kumar et al., 2019a; Chen et al., 2021). In this work, we want to leverage

suboptimal, reward-free data, such as play data or failed trajectories. In addition, we would like to avoid the additional complexity of annotating the data with rewards or training a value function which the offline RL methods rely on.

Several works have also addressed learning from action-free data, such as using inverse models (Torabi et al., 2018; Baker et al., 2022), latent action models (Edwards et al., 2019; Schmeckpeper et al., 2020; Bruce et al., 2024), or representation learning (Radosavovic et al., 2023; Wu et al., 2023b; Cui et al., 2024). In this work we focus on a simple recipe for robotic imitation learning that is naturally able to leverage action-free data through state forecasting.

**Diffusion and Image Prediction in Robot Learning.** Diffusion models, due to their expressivity and training and sampling stability, have been applied to robot learning tasks. Diffusion has been used in offline reinforcement learning (Hansen-Estruch et al., 2023) and imitation learning (Chi et al., 2023). Diffuser (Janner et al., 2022) learns a denoising diffusion model on trajectories, including both states and actions, in a model-based reinforcement learning setting. Decision Diffuser (Ajay et al., 2023) extends Diffuser by showing compositionality over skills, rewards, and constraints, and instead diffuses over states and uses an inverse dynamics model to extract actions from the plan. Due to the complexity of modeling image trajectories, Diffuser and Decision Diffuser restrict their applications to low-dimensional states.

To scale up to diffusing over higher-dimensional plans, UniPi (Du et al., 2023a; Ko et al., 2023) adapts video models for planning. Unlike works that rely on foundation models and video models for planning (Du et al., 2023b; Yang et al., 2024; Zhou et al., 2024), our method avoids computational and modeling complexities of generative video modeling by planning over latent embeddings instead.

Previous works have used world models to plan over images in a compact latent space (Hansen et al., 2024; Hafner et al., 2019; 2020). In contrast with these works, we focus on single task imitation instead of reinforcement learning.

Many prior works argue that state forecasting objectives are uniquely suitable for robotics to improve planning quality with trajectory optimization or reinforcement learning (Finn & Levine, 2017; Yang et al., 2023), by using the model directly to plan future states (Du et al., 2023b;a), as well as representation learning (Wu et al., 2023a; Radosavovic et al., 2023). We follow this line of work by proposing a planning-based method competitive to state-of-the-art robotic imitation learning that can leverage heterogeneous data sources.

## 3. Background

**Diffusion Models** Diffusion models, such as Denoising Diffusion Probabilistic Models (DDPMs), are likelihood-based generative models that learn an iterative denoising process from a Gaussian prior to a data distribution (Sohl-Dickstein et al., 2015; Ho et al., 2020; Song et al., 2020). During training time, DDPMs are trained to reverse a single noising step. Then, at sampling time, to reverse the diffusion process, the model iteratively denoises a sample drawn from the known Gaussian prior.

Diffusion models may also be conditioned on additional context. For example, text-to-image generative models are conditioned on text, Diffusion Policy is conditioned on visual observations, and Decision Diffuser can be conditioned on reward, skills, and constraints.

Recent generative models have used Latent Diffusion Models, which trains a diffusion model in a learned, compressed latent space (Rombach et al., 2022; Peebles & Xie, 2023; Blattmann et al., 2023) to improve computational and memory efficiency. The latent space is typically learned via an autoencoder, with encoder $\mathcal{E}$ and decoder $\mathcal{D}$ trained to reconstruct $x \approx \hat{x} = \mathcal{D}(\mathcal{E}(x))$. Instead of diffusing over $x$, the diffusion model is trained on diffusing over $\mathbf{z} = \mathcal{E}(x)$.

**Imitation Learning** In the imitation learning framework, we assume access to a dataset of expert demonstrations, $\mathcal{D} \triangleq \{(s_0, x_0, a_0), \ldots, (s_T, x_T, a_T)\}$, generated by $\pi_E$, an expert policy. $s_i, x_i, a_i$ correspond to the state, image, and action at timestep $i$ respectively. The imitation learning objective is to extract a policy $\hat{\pi}(a|s, x)$ that most closely imitates $\pi_E$. In robotics, this is typically approached through behavior cloning, which learns the mapping between states and actions directly via supervised learning. We consider single-task imitation, where the dataset corresponds to a single task.

Diffusion Policy (Chi et al., 2023) is an instantiation of diffusion models for imitation learning that has shown success in simulated and real-world tasks. Diffusion Policy uses a DDPM objective to model the distribution of action sequences, conditioned on observations. The CNN instantiation uses a Conditional U-Net Architecture, based on the 1D Temporal CNN in (Janner et al., 2022), which encourages temporal consistency due to the inductive biases of convolutions. LDP's planner architecture is based on the CNN-based Diffusion Policy, though we forecast latent states instead of actions.

Datasets of expert demonstrations often do not provide sufficient state distribution coverage to effectively solve a given task with imitation learning. However, there often exists additional data in the form of action-free or suboptimal data, which may consist of failed policy rollouts, play data, or miscellaneous environment interactions. Unfortunately,

behavior cloning assumes access to data annotated with optimal actions, so such additional data cannot be easily incorporated into training.

## 4. Latent Diffusion Planning

Latent Diffusion Planning consists of three parts, as shown in Figure 1: (1) Training an image encoder via an image reconstruction loss, (2) learning an inverse dynamics model to extract actions $a_t$ from pairs of latent states $\mathbf{z}_t, \mathbf{z}_{t+1}$, and (3) learning a planner to forecast future latents $\mathbf{z}_t$.

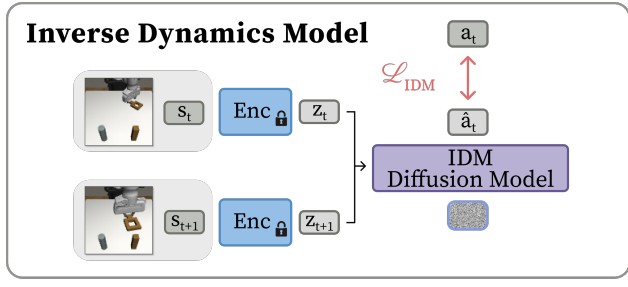

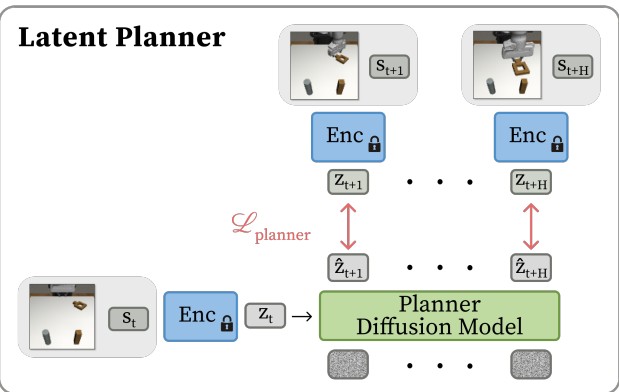

*Figure 2.* After training the encoder, Latent Diffusion Planning trains two diffusion models. *Top:* We train a inverse dynamics model (IDM) with a diffusion objective to directly extract the actions that will be used for control from pairs of latent states. *Bottom:* We train a powerful latent diffusion model to forecast a chunk of future latent states. The planner and the IDM are used together to produce an action chunk, similar to (Chi et al., 2023).

### 4.1. Learning the Latent Space

We circumvent planning over high-dimensional image observations by planning over a learned latent space. Similar to prior work in planning with world models (Watter et al., 2015; Ha & Schmidhuber, 2018; Hafner et al., 2020), we learn this latent space using an image reconstruction objective. Our planner thus becomes similar to video models that forecast image frames in a learned latent space (Yan et al., 2021; Hong et al., 2022; Blattmann et al., 2023).

In this work, we train a variational autoencoder (Kingma

---

**Algorithm 1** Inference with Latent Diffusion Planning

1: **Input:** Encoder $\mathcal{E}$, Planner $\epsilon_\psi$, IDM $\epsilon_\xi$, Planner Diffusion Timesteps $T_p$, IDM Diffusion Timesteps $T_{\text{IDM}}$, Planning Horizon $H_p$, Action Horizon $H_a$

2: Observe initial state $s_0$ and image $x_0$; $k = 0$
3: **while** not done **do**
4:     $\mathbf{z}_k \leftarrow (\mathcal{E}(x_k), s_k)$

    // Diffuse over latent embedding plan
5:     $\hat{\mathbf{z}}_{k+1}, ..., \hat{\mathbf{z}}_{k+H_p} \sim (0, I)$
6:     **for** $t = T_p \ldots 1$ **do**
7:         $\hat{\epsilon} \leftarrow \epsilon_\psi(\hat{\mathbf{z}}_{k+1}, ..., \hat{\mathbf{z}}_{k+H_p}; \mathbf{z}_k, t)$
8:         Update $\hat{\mathbf{z}}_{k+1}, ..., \hat{\mathbf{z}}_{k+H_p}$ using DDPM update with $\hat{\epsilon}$
9:     **end for**

    // Diffuse over actions between latent embeddings
10:    **for** $i = 0 \ldots H_a - 1$ **do**
11:       $\hat{a}_{k+i} \sim (0, I)$    // Predict action for each timestep in action horizon
12:       **for** $t = T_{\text{IDM}} \ldots 1$ **do**
13:          $\hat{\epsilon} \leftarrow \epsilon_\xi(\hat{a}_{k+i}; \hat{\mathbf{z}}_{k+i}, \hat{\mathbf{z}}_{k+i+1}, t)$
14:          Update $\hat{a}_{k+i}$ using DDPM update with $\hat{\epsilon}$
15:       **end for**
16:    **end for**

    // Execute actions
17:    **for** $i = 0 \ldots H_a - 1$ **do**
18:       $s_{k+i+1} \leftarrow \text{env.step}(s_{k+i}, \hat{a}_{k+i})$
19:    **end for**
20:    $k \leftarrow k + H_a$
21: **end while**

---

& Welling, 2014; Rezende et al., 2014) to obtain a latent encoder $\mathcal{E}$ and decoder $\mathcal{D}$. Specifically, we optimize the $\beta$-VAE (Higgins et al., 2017) objective, where $\mathbf{x}$ is our original image, $\mathbf{z}$ is our learned latent representation of the image, $\theta$ are the parameters for our decoder, $\phi$ are the parameters for our encoder, and $\beta$ is the weight for the KL regularization term:

$$\mathcal{L}_{\text{VAE}}(\theta, \phi; \mathbf{x}, \mathbf{z}, \beta) = \mathbb{E}_{q_\phi(\mathbf{z}\,|\,\mathbf{x})}[\log p_\theta(\mathbf{x}\,|\,\mathbf{z})] \\ - \beta \mathcal{D}_{\text{KL}}(q_\phi(\mathbf{z}\,|\,\mathbf{x})||p(\mathbf{z})) \quad (1)$$

In practical scenarios, we may have a limited expert demonstration dataset, but much larger and diverse suboptimal or action-free datasets. In this phase of learning, we can make use of the visual information in such datasets for training a more robust latent encoder.

## 4.2. Planner and Inverse Dynamics Model

Our policy consists of two separate modules: (1) a planner over latent embeddings $\mathbf{z}$, and (2) an inverse dynamics model similarly operating over the latent embeddings. The planner and IDM are both parameterized as DDPM models, motivated by the expressivity that diffusion models offer.

The planner is conditioned on the current latent embedding, which consists of the concatenated latent image embedding and robot proprioception, and diffuses over a horizon of future embeddings. We use Diffusion Policy's Conditional U-Net architecture, with a CNN backbone. Concretely, we optimize the following objective:

$$\mathcal{L}_{\text{planner}}(\psi, \mathbf{z}) = \mathbb{E}_{t,\epsilon}[||\epsilon_\psi(\hat{\mathbf{z}}_{k+1}, \ldots, \hat{\mathbf{z}}_{k+H}; \mathbf{z}_k, t) - \epsilon||^2] \tag{2}$$

where $\mathbf{z}_k$ is the latent embedding at timestep $k$ of the trajectory; $\hat{\mathbf{z}}_{k+1}, \ldots, \hat{\mathbf{z}}_{k+H}$ is the noised latent embedding sequence, with corresponding noise $\epsilon$; $H$ is the maximum horizon of the forecasted latent plan; $t$ is the diffusion noise timestep; and $\psi$ are the parameters of the planner diffusion model.

Our inverse dynamics model is trained to reconstruct the action between a pair of states, conditioned on their associated latent embeddings. We use the MLPResNet architecture from IDQL (Hansen-Estruch et al., 2023) to diffuse actions, as it is more lightweight. We optimize the loss:

$$\mathcal{L}_{\text{IDM}}(\xi, \mathbf{z}) = \mathbb{E}_{t,\epsilon}[||\epsilon_\xi(\hat{a}_k; \mathbf{z}_k, \mathbf{z}_{k+1}, t) - \epsilon||^2] \tag{3}$$

where $\mathbf{z}_k$ is the latent embedding at timestep $k$ of the trajectory; $\hat{a}_k$ is the noised action, with corresponding noise $\epsilon$; $t$ is the diffusion noise timestep; and $\xi$ are the parameters of the inverse dynamics diffusion model.

Because our latent embedding is frozen from the learned VAE, the planner and IDM do not share parameters and can be trained separately (Figure 2). Then, at inference time, the two modules are combined to extract action sequences. First, the planner forecasts a future horizon of states via DDPM sampling (Alg. 1 Lines 6-9). Then, we use the inverse dynamics model to extract actions from latent embedding pairs produced by the planner, also via DDPM Diffusion (Alg. 1 Lines 12-15). Like Diffusion Policy, we employ receding-horizon control (Mayne & Michalska, 1988), and execute for a shorter horizon than the full forecasted horizon (Alg. 1 Lines 17-19).

## 5. Experiments

We seek to answer the following questions:

- Does Latent Diffusion Planning leverage *action-free* data for improved planning?

- Is Latent Diffusion Planning comparable to state-of-the-art imitation learning algorithms that leverage *suboptimal* data?

- Can Latent Diffusion Planning be an effective imitation learning method in a real-world robotics system, where there may be suboptimal or action-free data?

### 5.1. Experimental Setup

**Simulated Tasks** We focus our experiments on 4 image-based imitation learning tasks: (1) Robomimic Lift, (2) Robomimic Can, (3) Robomimic Square, and (4) ALOHA Sim Transfer Cube. Robomimic (Mandlekar et al., 2021) is a robotic manipulation and imitation benchmark, including the tasks Lift, Can, and Square. The Transfer Cube task is a simulated bimanual ALOHA task, in which one ViperX 6-DoF arm grabs a block and transfers it to the other arm (Zhao et al., 2023).

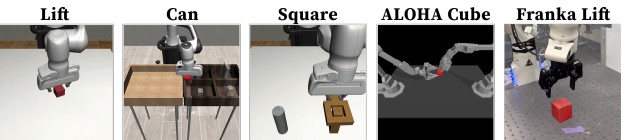

To demonstrate the effectiveness of Latent Diffusion Planning, we assume a low demonstration data regime, such that additional suboptimal or action-free data can improve performance. For Can and Square, we use 100 out of the 200 demonstrations in the Robomimic datasets; for Lift, we use 3 demonstrations out of the 200 total; and for Transfer Cube, we use 25 demonstrations. To further emphasize the importance of suboptimal data, these demonstrations cover a limited state space of the environment. Our suboptimal data consists of 500 failed trajectories from an undertrained behavior cloning agent. Our action-free data consists of 100 demonstrations for Lift, Can, and Square from the Robomimic dataset, and 25 demonstrations for Cube. We evaluate the success rate out of 50 trials, using the best checkpoint from the last 5 saved checkpoints, with 2 seeds.

**Real World Task** We create a real world implementation of the Robomimic Lift task, where the task is to pick up a red block from a randomly initialized position. We use a Frank Panda 7 degree of freedom robot arm, with a wrist-mounted Zed camera. We use the DROID setup (Khazatsky et al., 2024) and teleoperate via the Oculus Quest 2 headset. We use cartesian pose control.

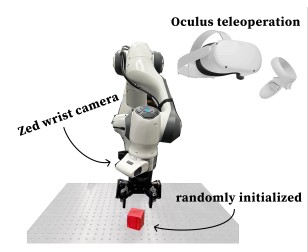

*Table 1.* **Leveraging Action-Free Data.** LDP is able to leverage suboptimal and action-free data. We compare against DP baselines that leverage action-free data by using an IDM to relabel actions (DP-VPT) and video planning models that may use action-free data for the video planner (UniPi). We find LDP to perform better than both approaches, especially when combined with suboptimal data.

| Method | Lift | Can | Square | ALOHA Cube | Average |
|---|---|---|---|---|---|
| DP | $0.60 \pm 0.00$ | $0.63 \pm 0.01$ | $0.48 \pm 0.00$ | $0.32 \pm 0.00$ | 0.51 |
| DP-VPT | $0.69 \pm 0.01$ | $0.75 \pm 0.01$ | $0.48 \pm 0.04$ | $0.45 \pm 0.03$ | 0.59 |
| UniPi-OL + Action-Free | $0.09 \pm 0.05$ | $0.23 \pm 0.03$ | $0.07 \pm 0.03$ | $0.02 \pm 0.00$ | 0.11 |
| UniPi-CL + Action-Free | $0.14 \pm 0.02$ | $0.32 \pm 0.04$ | $0.09 \pm 0.01$ | $0.17 \pm 0.03$ | 0.18 |
| LDP | $0.69 \pm 0.03$ | $0.70 \pm 0.02$ | $0.46 \pm 0.00$ | $0.64 \pm 0.04$ | 0.65 |
| LDP + Action-Free | $0.67 \pm 0.01$ | $0.78 \pm 0.04$ | $0.47 \pm 0.03$ | $0.70 \pm 0.02$ | 0.66 |
| LDP + Action-Free + Subopt | $\mathbf{1.00} \pm 0.00$ | $\mathbf{0.98} \pm 0.00$ | $\mathbf{0.83} \pm 0.01$ | $\mathbf{0.97} \pm 0.01$ | **0.95** |

We collect 82 demonstrations, collect 84 suboptimal trajectories, and 12 action-free demonstrations.

Our suboptimal data consists of failed trajectories from policy evaluations. This is an effective way to reuse the data generated during iterations of training, that algorithms modeling actions, such as Diffusion Policy, cannot use. Action-Free data may consist of kinesthetic demonstrations, human videos, or handheld demonstrations (Chi et al., 2024). In our case, we collect teleoperated demonstrations with actions removed.

To evaluate our policies, we calculate the success rate across 45 evaluation trials. To thoroughly evaluate performance across the initial state space, we evaluate across a grid of 3x3 points, with 5 attempts per point. We evaluate 3 seeds per method.

**Baselines** We consider two main categories of baselines: (1) Imitation learning with suboptimal or action-free data (DP, RC-DP, DP+Repr, DP PT + FT, DP-VPT), and (2) Video planning (UniPi-OL, UniPi-CL).

- **Diffusion Policy** (**DP**) is a state-of-the-art imitation learning algorithm.

- **Reward-Conditioned Diffusion Policy** (**RC-DP**) utilizes suboptimal actions by conditioning the policy on a binary value indicating whether the action chunk comes from optimal demonstrations or not. This method is inspired by reward-conditioned approaches (Kumar et al., 2019b; Chen et al., 2021).

- **Diffusion Policy with Representation Learning** (**DP+Repr**) uses a VAE pretrained on demonstration, suboptimal, and action-free data as the observation encoder. This is representative of methods that leverage suboptimal data through representation learning.

- **Diffusion Policy Pretrain + Finetune (DP PT + FT)** pretrains on suboptimal trajectories and finetunes on demos. This is representative of methods that leverage suboptimal data through learning trajectory-level features.

- **Diffusion Policy with Video PreTraining (DP-VPT)** trains an inverse dynamics model to relabel action-free data, inspired by VPT (Baker et al., 2022) and BCO (Torabi et al., 2018).

- **Open-Loop UniPi** (**UniPi-OL**) is based off of UniPi (Du et al., 2023a), a video planner for robot manipulation. UniPi-OL generates a single video trajectory, extracts actions, and executes the actions in an open-loop fashion. We use a goal-conditioned behavior cloning agent to reach generated subgoals (Wen et al., 2024).

- **Closed-Loop UniPi** (**UniPi-CL**) is a modification that allows UniPi to perform closed-loop replanning over image chunks. Like LDP, UniPi-CL generates dense plans instead of waypoints, though in image space. We learn an inverse dynamics model to extract actions.

### 5.2. Imitation Learning with Action-Free Data

In Table 1, we examine how action-free data can be used to improve imitation learning policies. Imitation learning policies that model actions, such as DP, are unable to natively use action-free data, while planning-based approaches can benefit from this additional data.

One approach is to relabel action-free data using an inverse dynamics model. We find this to be effective for most tasks, showing that the reannotated actions are useful for policy improvement. However, we see that LDP is better able to leverage action-free data by directly using it for the planner, rather than generating possibly inaccurate actions to subsequently learn from.

Like LDP, video planning methods can directly use action-free data for improving the planner, which for UniPi is the

*Table 2.* **Leveraging Suboptimal Data.** Latent Diffusion Planning can utilize suboptimal data for improved performance across a suite of tasks. Video planning methods like UniPi consistently struggle, whereas reward-conditioned or pretraining DP approaches are able to improve DP policy performance. LDP is best able to use suboptimal data, especially when combined with action-free data.

| Method | Lift | Can | Square | ALOHA Cube | Average |
|---|---|---|---|---|---|
| DP | $0.60 \pm 0.00$ | $0.63 \pm 0.01$ | $0.48 \pm 0.00$ | $0.32 \pm 0.00$ | 0.51 |
| RC-DP | $0.40 \pm 0.04$ | $0.73 \pm 0.03$ | $0.66 \pm 0.02$ | $0.60 \pm 0.04$ | 0.60 |
| DP+Repr | $0.66 \pm 0.04$ | $0.61 \pm 0.01$ | $0.44 \pm 0.02$ | $0.25 \pm 0.03$ | 0.49 |
| DP PT + FT | $0.52 \pm 0.02$ | $0.67 \pm 0.01$ | $0.57 \pm 0.03$ | $0.78 \pm 0.00$ | 0.64 |
| UniPi-OL | $0.12 \pm 0.06$ | $0.28 \pm 0.02$ | $0.07 \pm 0.01$ | $0.00 \pm 0.00$ | 0.12 |
| UniPi-CL | $0.12 \pm 0.02$ | $0.30 \pm 0.02$ | $0.10 \pm 0.04$ | $0.15 \pm 0.07$ | 0.17 |
| LDP | $0.69 \pm 0.03$ | $0.70 \pm 0.02$ | $0.46 \pm 0.00$ | $0.64 \pm 0.04$ | 0.65 |
| LDP + Subopt | $0.84 \pm 0.06$ | $0.68 \pm 0.02$ | $0.55 \pm 0.03$ | $0.71 \pm 0.03$ | 0.70 |
| LDP + Action-Free + Subopt | $\mathbf{1.00} \pm 0.00$ | $\mathbf{0.98} \pm 0.00$ | $\mathbf{0.83} \pm 0.01$ | $\mathbf{0.97} \pm 0.01$ | **0.95** |

video generation model. We see limited improvement when using action-free data, and both DP and LDP still strongly outperform UniPi-CL and UniPi-OL. UniPi methods particularly struggle with more complex tasks like Square and ALOHA Cube, implying that video planning is still ineffective for tasks that require more precise manipulation skills.

Qualitatively, for UniPi-OL, we find that while the goal-conditioned agent is able to follow goals effectively, the policy still struggles with the difficult parts of the task, such as grasping the object. Forecasting goals does not provide the dense supervision for exactly how to grasp an object, and furthermore, UniPi-OL does not support replanning when a grasp is missed. UniPi-CL is able to address this by dense image forecasting, and consistently outperforms UniPi-OL. However, this closed-loop method is not only slow, but faces issues with video generation, such as regenerating static frames during parts of the task with less movement, leading the agent to be stuck in certain positions. This is especially noticeable for the ALOHA Cube task, where the agent is often stuck right before picking up the cube. Compared to UniPi-OL and UniPi-CL, LDP is able to circumvent many of these issues due to its latent planning and dense forecasting.

### 5.3. Imitation Learning with Suboptimal Data

In Table 2, we present imitation learning results with suboptimal data. First, LDP outperforms DP, which can only utilize data with optimal actions. We notice, especially, that LDP with suboptimal data typically improves further upon LDP, showing the potential of leveraging diverse data sources outside of the demonstration dataset.

Next, RC-DP, a conditional variant of DP that utilizes suboptimal data, outperforms DP. By learning from suboptimal data, RC-DP can learn priors of robot motions while distinguishing optimal action sequences. We hypothesize that for the Can, Square, and ALOHA Cube tasks, the primitive

motions of reaching toward or grasping the object, which are partially covered by the suboptimal dataset, provides a useful visuomotor prior for the policy. ALOHA Cube sees significant improvement, possibly because the larger action space of bimanual control benefits from additional reward-labelled data.

Next, we explore using suboptimal data for feature-learning. DP + Repr uses suboptimal data for pretraining a vision-encoder, and DP PT + FT for pretraining the visuomotor policy. DP + Repr only improves policy performance for the Lift task, implying that end-to-end training of the vision encoder learns stronger features for complex manipulation tasks. DP PT + FT is particularly successful for tasks that require more precise manipulation, such as Square and ALOHA Cube, implying that learned prior motions is a useful policy initialization. Both the success of RC-DP and DP PT + FT suggest that leveraging the suboptimal trajectories is a useful way to improve imitation learning results. LDP, which leverages suboptimal data for the latent encoder and IDM, has higher overall performance than these methods, averaging across the suite of simulated tasks.

Next, we compare against UniPi, which plans over image subgoals (OL) or image chunks (CL). Due to the low demonstration data regime, learning effective and accurate video policies is difficult, and LDP strongly outperforms UniPi-OL and UniPi-CL. In addition, we notice that UniPi-CL outperforms UniPi-OL for all tasks, implying that dense forecasting, even within the image domain, is more effective than goal-conditioned methods.

Finally, we find that LDP with action-free *and* suboptimal data leads to the strongest performance. We find a significant improvement in all of the simulated tasks, including compared to the variants of LDP that only use either action-free or suboptimal data. This suggests that the recipe for combining these two data sources can lead to a stronger

*Table 3.* **Real World Results** For the Franka Lift task, we find that LDP consistently outperforms DP. Suboptimal and action-free data, that DP is unable to use, further improves performance.

| Method | Success Rate |
|---|---|
| DP | $69.6 \pm 4$ |
| LDP | $73.3 \pm 7$ |
| LDP + Subopt | $74.8 \pm 4$ |
| LDP + Action-Free | $\mathbf{79.3} \pm \mathbf{3}$ |

planner and a more robust inverse dynamics model, leading to the best performing model for these tasks.

### 5.4. Imitation Learning in the Real World

Real world data is more expensive and time-consuming to collect; hence, examining the effect of easier-to-collect suboptimal and action-free data provides insights for scalable learning. In Table 3, we provide results on a Franka Lift Cube task. In this task, we examine the performance of DP, which can only leverage action-labeled data, with our method, which can use suboptimal and action-free data.

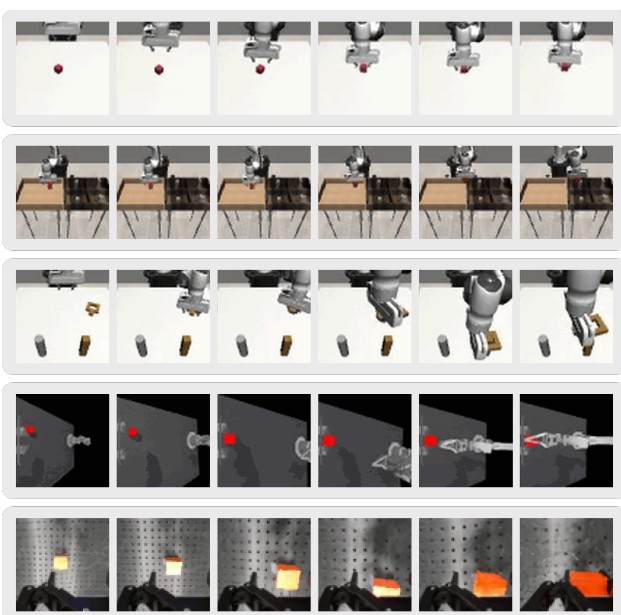

*Figure 3.* **Visualizations of Generated Plans.** LDP produces dense, closed-loop plans. Here, we visualize decoded latents selected from an LDP trajectory for the Lift, Can, Square, ALOHA Cube, and Franka Lift tasks.

We find that LDP is able to consistently outperform DP, especially with the addition of action-free data. For real-world systems, this is promising, as collecting high-quality demos can be difficult and time intensive, whereas utilizing suboptimal trajectories, often a byproduct of evaluating policies, or collecting action-free trajectories more efficiently, such

as in (Chi et al., 2024), can be a scalable direction.

### 5.5. Ablation: LDP Hierarchical

In an attempt to understand the effect of LDP's dense forecasting, we compare LDP with a hierarchical version of LDP (LDP Hierarchical). In this implementation, LDP Hierarchical plans over subgoals 4 steps apart, and extracts 4 actions between pairs of forecasted latent states. Thus, LDP Hierarchical is closed-loop, yet the planner operates at a slightly abstracted level compared to non-hierarchical LDP.

In Table 4, we find that LDP outperforms LDP Hierarchical across the 3 Robomimic tasks. This suggests that dense forecasting is an important part and contribution of LDP. This also reflects UniPi results in Table 1 and Table 2 that show closed-loop planning outperforms open-loop planning.

*Table 4.* **Hierarchical Ablation**. LDP's dense forecasting outperforms a hierarchical variant of LDP.

| Method | Lift | Can | Square |
|---|---|---|---|
| LDP | $0.69 \pm 0.03$ | $0.70 \pm 0.02$ | $0.46 \pm 0.00$ |
| LDP Hier. | $0.53 \pm 0.03$ | $0.62 \pm 0.04$ | $0.31 \pm 0.01$ |
| LDP + Subopt | $0.84 \pm 0.06$ | $0.68 \pm 0.02$ | $0.55 \pm 0.03$ |
| LDP Hier. + Subopt | $0.65 \pm 0.03$ | $0.60 \pm 0.02$ | $0.43 \pm 0.05$ |

## 6. Discussion

We presented Latent Diffusion Planning, a simple planning-based method for imitation learning. We show that our design using powerful diffusion models for latent state forecasting enables competitive performance with state-of-the-art imitation learning. We further show this latent state forecasting objective enables us to easily leverage heterogeneous data sources. In the low-demonstration data imitation regime, LDP outperforms prior imitation learning work that does not leverage such additional data as effectively.

**Limitations.** One limitation of the current approach is that the latent space for planning is simply learned with a variational autoencoder and might not learn the most useful features for control. Future work will explore different representation learning objectives. Further, our method requires diffusing over states, which incurs additional computational overhead as compared to diffusing actions. However, we expect continued improvements in hardware and inference speed will mitigate this drawback. Finally, we did not explore applying recent improvements in diffusion models (Peebles & Xie, 2023; Lipman et al., 2022), which may be important in large data regimes.

**Future work.** We have validated in simulation and real the hypothesis that latent state forecasting can leverage heterogeneous data sources. Future work can also evaluate whether this can be used to further improve more complex

real-world tasks. One direction is to use a diverse dataset of human collected data, such as with handheld data collection tools (Young et al., 2021). Another approach would be to use autonomously collected robotic data (Bousmalis* et al., 2023). As these alternative data sources are easier to collect than demonstrations, they represent a different scaling paradigm that can outperform pure behavior cloning approaches. By presenting a method that can leverage such data, we believe this work makes a step toward more performant and general robot policies.

## Impact Statement

This paper presents work whose goal is to advance the field of Machine Learning. There are many potential societal consequences of our work, none which we feel must be specifically highlighted here.

## Acknowledgments

We would like to thank members of ILIAD and IRIS at Stanford for fruitful discussions and feedback. We are grateful to Archit Sharma, Annie Chen, Austin Patel, Hengyuan Hu, Jensen Gao, Joey Hejna, Mallika Parulekar, Mengda Xu, Philippe Hansen-Estruch, Shuang Li, Suvir Mirchandani, and Zeyi Liu for helpful conversations.

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

# A. Appendix

## A.1. Additional Ablations

### A.1.1. LDP WITH PRETRAINED EMBEDDINGS

We evaluate the effectiveness of planning and extracting actions from pretrained embeddings with a DINOv2 ablation. In this approach, we replace our VAE embeddings with DINOv2 embeddings, with no other changes to the LDP architecture.

The DINOv2 latent embedding is 384-dimensional. We found directly planning over the DINOv2 embeddings does not lead to good learned behaviors (0% success), which we hypothesize is due to the challenges of planning over a large latent space. Thus, as an alternative, we fix a random projection matrix, reducing the 384 dim feature space to 16 dim, matching LDP. We choose this, because this is a straightforward way of using frozen embeddings. It may be possible to plan over the large embeddings space with a much larger and complex model, or it may be possible to learn alternative ways to project DINOv2 embeddings to a lower-dimensional space, but this may lead to fundamental changes in our method, so we leave more complicated approaches to use pretrained embeddings for future work. We include results on Robomimic tasks in Table 5.

*Table 5.* **DINOv2 Ablation**. Planning over frozen embeddings (DINOv2), even when projected to a lower dimensional space, has significantly lower success rate than planning over VAE embeddings. This suggests that the learned latent space is crucial to LDP's performance, and that a compact, well-structured latent space leads to optimal performance.

| Method | Latent Dim | Lift | Can | Square |
|---|---|---|---|---|
| LDP w/ DINOv2 (frozen) | 384 | $0.00 \pm 0.00$ | $0.00 \pm 0.00$ | $0.00 \pm 0.00$ |
| LDP w/ DINOv2 (frozen, randomly projected) | 16 | $0.44 \pm 0.24$ | $0.03 \pm 0.01$ | $0.01 \pm 0.01$ |
| LDP + Action-Free + Subopt | 16 | $1.00 \pm 0.00$ | $0.98 \pm 0.00$ | $0.83 \pm 0.01$ |

### A.1.2. UNIPI FROM PRETRAINED MODEL

In our experimental results, we train our UniPi model from scratch on demonstration videos. To provide a comparison to training UniPi from pre-trained weights, we use the AVDC-THOR pretrained checkpoint (Ko et al., 2023). We finetune for an additional 50k steps at learning rate 1e-4.

In Table 6 and Table 7, we find that the pretrained UniPi model does not substantially improve performance, likely because the pretrained model quality is limited. For UniPi-OL, there appears to be slightly improvement when pretraining from scratch, possibly because without closed-loop planning, the pretrained initialization may be helpful.

*Table 6.* **UniPi from Pretrained Checkpoint**

| Method | Lift | Can | Square |
|---|---|---|---|
| UniPi-OL (from scratch) | $0.09 \pm 0.03$ | $0.27 \pm 0.01$ | $0.07 \pm 0.01$ |
| UniPi-OL (from pretrain) | $0.13 \pm 0.05$ | $0.31 \pm 0.01$ | $0.07 \pm 0.01$ |
| UniPi-CL (from scratch) | $0.12 \pm 0.02$ | $0.30 \pm 0.02$ | $0.10 \pm 0.04$ |
| UniPi-CL (from pretrain) | $0.13 \pm 0.05$ | $0.23 \pm 0.01$ | $0.08 \pm 0.02$ |

*Table 7.* **UniPi + Action-Free from Pretrained Checkpoint**

| Method | Lift | Can | Square |
|---|---|---|---|
| UniPi-OL (from scratch) | $0.11 \pm 0.03$ | $0.25 \pm 0.03$ | $0.05 \pm 0.03$ |
| UniPi-OL (from pretrain) | $0.10 \pm 0.08$ | $0.26 \pm 0.00$ | $0.08 \pm 0.02$ |
| UniPi-CL (from scratch) | $0.14 \pm 0.02$ | $0.32 \pm 0.04$ | $0.09 \pm 0.01$ |
| UniPi-CL (from pretrain) | $0.11 \pm 0.03$ | $0.25 \pm 0.01$ | $0.10 \pm 0.00$ |

## A.2. Implementation Details

**Diffusion Policy** We use a Jax reimplementation of the convolutional Diffusion Policy, which we verify can reproduce reported Robomimic benchmark results.

**UniPi** We use the open-source implementation of UniPi (Ko et al., 2023). For UniPi-OL and UniPi-CL, we predict 7 future frames. During training time, for UniPi-OL, the 7 future frames are evenly sampled from a training demonstration. For UniPi-CL, the 7 future frames are the next consecutive frames.

The goal-conditioned behavior cloning agent is implemented as a goal-conditioned Diffusion Policy (Chi et al., 2023) agent, and it is trained on chunks of 16. The inverse dynamics model is based off of IDQL (Hansen-Estruch et al., 2023), and shares the same architecture as the IDM used in LDP.

We train the video prediction models for 200k gradient steps with batch size 16.

**LDP** The LDP VAE is adapted from Diffusion Transformer (Peebles & Xie, 2023). The planner is based directly off of the convolutional U-Net from Diffusion Policy (Chi et al., 2023), with modifications to plan across latent embeddings instead of action chunks. The IDM is based off of IDQL (Hansen-Estruch et al., 2023).

*Table 8.* Diffusion Policy Architecture Hyperparameters

|  | UniPi-OL GCBC | DP and LDP | LDP - ALOHA Cube |
|---|---|---|---|
| down_dims | [256, 512, 1024] | [256, 512, 1024] | [512, 1024, 2048] |
| n_diffusion_steps | 100 | 100 | 100 |
| batch_size | 256 | 256 | 256 |
| lr | 1e-4 | 1e-4 | 1e-4 |
| n_grad_steps | 500k | 500k | 500k |

*Table 9.* IDM Architecture Hyperparameters

|  | UniPi-CL IDM | LDP IDM |
|---|---|---|
| n_blocks | 3 | 3 (Lift, Square, ALOHA Cube); 5 (Can) |
| n_diffusion_steps | 100 | 100 |
| batch_size | 256 | 256 |
| lr | 1e-4 | 1e-4 |
| n_grad_steps | 500k | 500k |

*Table 10.* VAE Architecture Hyperparameters

|  | VAE |
|---|---|
| block_out_channels | [128, 256, 256, 256, 256, 256] |
| down_block_types | [DownEncoderBlock2D] x6 |
| up_block_types | [UpDecoderBlock2D] x6 |
| latent_channels | 4 |
| Latent Dim | (2, 2, 4) |
| Lift KL Beta | 1e-5 |
| Can KL Beta | 1e-6 |
| Square KL Beta | 1e-6 |
| ALOHA Cube KL Beta | 1e-7 |
| n_grad_steps | 300k |

### A.3. Simulation Experiments

#### A.3.1. ENVIRONMENT

For all environments, we use an observation horizon of 1. For Robomimic tasks, we use the third-person view image. For ALOHA Cube and Franka Lift, we use the wrist camera. All images are 64x64.

### A.3.2. DATASET SIZES

We detail the design decisions for the number of expert, action-free, and suboptimal trajectories. For simulated robotics tasks, we kept consistent the action-free and suboptimal trajectories: 100 action-free trajectories for robomimic tasks and 500 suboptimal trajectories for all tasks.

**Number of Demonstrations** Robomimic includes 200 proficient demonstrations per task, and ALOHA includes 50 demonstrations per task. To showcase the effectiveness of our method, we always take half the number of demonstrations– 100 for Robomimic tasks, and 25 for ALOHA tasks. This is a consistent protocol we use to select the number of demonstrations. However, Lift is a particularly simple task, so in order to not saturate our baseline methods with excessive numbers of demos, we reduce the number of demonstrations to 3. Without this adjustment, Lift results would not be useful to discern the effect of different types of data, and furthermore, Lift results would all be saturated.

**Number of Action-Free Trajectories** We chose to use the remaining half of the demonstrations as action-free trajectories. Hence, ALOHA uses 25 action-free trajectories, and Robomimic uses 100 action-free trajectories. Note again that Lift only used 3 expert demonstrations, but in order to keep our numbers as consistent as possible and avoid further confusion, we opted to use 100 action-free trajectories.

### A.3.3. METHOD DETAILS

**LDP VAE** We train with additional action-free and suboptimal data. This VAE is shared across a single task for all variants of LDP, and including for DP + Repr.

**LDP + Subopt** We train the inverse dynamics model with batches of 50% optimal, 50% suboptimal data.

**LDP + Action-Free** We train only the planner on additional action-free demos. The IDM is trained only on expert demonstrations.

**LDP + Action-Free + Subopt** We train only the planner on additional action-free demos. We train the inverse dynamics model with batches of 50% optimal, 50% suboptimal data.

**LDP Hierarchical** The IDM is parameterized as a Conditional U-Net, like the planner, but the IDM is smaller and has down_dims=[256, 512].

**LDP DINOv2** In this variant, we plan over pretrained DINOv2 (Oquab et al., 2023; Darcet et al., 2023) embeddings instead of VAE embeddings. We use the ViT-S variant and pad the 64x64 image to be 70x70, which matches the 14x14 patch size. The batch size is reduced to 128, but the remaining hyperparameters are identical to LDP. We further project the 384-dimensional embedding to 16-dimensional via a random matrix, in order for easier planning. Results are presented in Appendix A.1.1

**DP + Repr** We use the same VAE that LDP methods used.

**DP PT + FT** We pretrain for 300k steps and finetune with a lower learning rate (1e-5) for 200k steps.

**DP-VPT** We use an inverse dynamics model trained for 500k steps on curated data, as used in LDP, to relabel actions for the action-free data.

**UniPi** We train two goal-conditioned agents and two inverse dynamics models and report the average success. For UniPi-OL evaluations, we predetermine the number of steps for the GCBC to reach each image subgoal based on the demonstration lengths. For Lift, evaluation episode lengths are 60 steps; Can is 140 steps; and Square is 160 steps. This maximum horizon is also enforced for UniPi-CL evaluations, for consistency.

