# OpenReview forum: "Latent Diffusion Planning for Imitation Learning"
_ICML.cc/2025/Conference — ICML 2025 spotlightposter_

### Official Review · Reviewer_LWMg · 2025-03-09

**Overall Recommendation:** 4

**Summary:**

The paper proposes latent diffusion planning (LDP), a method for imitation learning featuring 3 components: 1) A variational autoencoder, mapping images to a latent spaces 2) A latent diffusion planner, which generates a sequence of latent states that the policy should visit 3) An inverse dynamics model, also leveraging diffusion, which associates an action to a latent transition. The interesting property of this framework is that it can use suboptimal demonstrations (with actions) to refine the inverse dynamics model, and unlabeled expert videos to improve the latent planner. Experiments show that this setup better makes use of unlabeled or suboptimal demonstrations than previous methods (on average).

**Claims And Evidence:**

The main claims of the papers are 1) LDP outperforms previous methods thanks to its better usage of unlabeled and suboptimal data 2) LDP can be applied to real robots, where collecting data with actions can be costly.

The results listed in Table 1 and 2 are convincing. LDP is competitive with the best baselines in all the considered tasks, often outperforming them. The authors make the effort of fairly comparing LDP with other methods by providing them with the same data LDP has access to. For example, relabeling action-free data with an inverse dynamics model is a strong alternative to LDP (which in fact performs comparably to LDP in Lift and Square), but overall LDP is stronger.

I would suggest removing LDP + Subopt from the first table, or to merge the 2, because I found it confusing at a first read (to my understanding, the baselines in table 1 do not use suboptimal data).

Experiments using Franka also show an improvement over DP, especially using action-free data.

One possible issue in using LDP is that often the performance improvement is unpredictable - in some cases using action-free or suboptimal trajectories leads to a large improvement, sometimes even to a small decrease in performance. Understanding in which situations suboptimal or action-free data are beneficial would improve the applicability of the method.

**Essential References Not Discussed:**

The background discussion is broad.

**Experimental Designs Or Analyses:**

I do not see particular issues with the experimental design. My only concern is the lack of justification for the choice of labeled / unlabeled / suboptimal subsets, which might have a relevant impact on the results.

**Methods And Evaluation Criteria:**

Yes, the benchmarks correctly support the claims. Additional experiments which would help understanding the role of action-free and suboptimal data could evaluate their role more precisely. For example, the authors could show a graph of the performance (maybe in one or in a subset of tasks) as a function of the amount of suboptimal / action-free data, to better understand whether their impact saturates at a certain point, and how many labeled optimal examples are necessary for the method to be effective.

**Other Comments Or Suggestions:**

108: which enabling
252 right column - the paragraph on the number of labeled trajectories and suboptimal / action free is difficult to follow, a small table would be clearer
294: remove use or collect

**Other Strengths And Weaknesses:**

Strenghts:
* Clarity of exposition
* Good comparison with baselines
* Elegant architecture
* Convincing results

Weaknesses:
* Lack of evaluation of how many labeled / unlabeled / suboptimal trajectories are needed for optimal performance

**Questions For Authors:**

- Does using action-free and suboptimal trajectories at the same time further improve the results?
- Does training the inverse dynamics on suboptimal trajectories, that will likely never be encountered during deployment (the planner will try to follow the optimal trajectories) actually improve action regression?
- Are the action-free data always from expert policies? To my understanding, suboptimal action-free trajectories cannot be used because they would decrease the performance of the planner.

**Relation To Broader Scientific Literature:**

The method is compared to strong baselines and therefore its relevance in the context of the broader literature is clear.

**Theoretical Claims:**

The paper makes no theoretical claims.

---

> ### Author Rebuttal · Authors · 2025-04-01
>
> Dear Reviewer,
>
> Thank you for your detailed feedback on our project. We are happy that you found the paper to be well-written and clear. We address your questions and comments below. Please let us know whether there are any other concerns you have that prevent you from increasing your score.
>
> **Q1: Choice of labeled / unlabeled / suboptimal trajectories are needed for optimal performance**
>
> For different robotics tasks, we tried to keep most of the parameters the same: 100 action-free trajectories for robomimic tasks and 500 suboptimal trajectories for all tasks. However, we chose:
>
> \# of demonstrations: Robomimic, by default, includes 200 proficient demonstrations per task, and ALOHA includes 50 demonstrations per task. To showcase the effectiveness of our method when the number of demonstrations is low, we chose to take **half** the number of demonstrations-- 100 for Robomimic tasks, and 25 for ALOHA tasks. However, Lift is a very simple task, and even with around 10 demonstrations, behavior cloning policies can achieve very high accuracy. Thus, specifically due to the simplicity of the Lift task, we chose 3 demonstrations to adhere to the low demonstration setting.
>
> \# of action-free trajectories: we chose to use the remaining **half** of the demonstrations as action-free trajectories. This meant 100 demonstrations for Robomimic tasks, and 25 demonstrations for ALOHA. Since Lift only used 3 demonstrations, we considered using only 3 action-free trajectories. However, because we wanted to try and remain consistent with the number of trajectories, we chose to use 100 action-free demonstrations.
>
> **Q2: Does using action-free and suboptimal trajectories at the same time further improve the results?**
> Yes, using action-free and suboptimal trajectories further improves results and strongly exceeds our baselines. We have included updated results. Here is a how LDP + Action-Free + Suboptimal compares against just LDP without the additional trajectories:
>
> In addition, we have reran UniPi with identical hyperparameters to LDP, per Reviewer PyaC’s suggestion.
>
> | Method                     | Lift         | Can          | Square       | ALOHA Cube   | Average |
> |----------------------------|--------------|--------------|--------------|--------------|---------|
> | DP                         | 0.60 +- 0.00 | 0.63 +- 0.01 | 0.48 +- 0.00 | 0.32 +- 0.00 | 0.51    |
> | DP-VPT                     | 0.69 +- 0.01 | 0.75 +- 0.01 | 0.48 +- 0.04 | 0.45 +- 0.03 | 0.59    |
> | UniPi-OL + Action-Free     | 0.09 +- 0.05 | 0.23 +- 0.03 | 0.07 +- 0.03 | 0.02 +- 0.00 | 0.11    |
> | UniPi-CL + Action-Free     | 0.14 +- 0.02 | 0.32 +- 0.04 | 0.09 +- 0.01 | 0.17 +- 0.03 | 0.18    |
> | LDP                        | 0.69 +- 0.03 | 0.70 +- 0.02 | 0.46 +- 0.00 | 0.64 +- 0.04 | 0.65    |
> | LDP + Action-Free          | 0.67 +- 0.01 | 0.78 +- 0.04 | 0.47 +- 0.03 | 0.70 +- 0.02 | 0.66    |
> | LDP + Action-Free + Subopt | 1.00 +- 0.00 | 0.98 +- 0.00 | 0.83 +- 0.01 | 0.97 +- 0.01 | 0.95    |
>
> | Method                     | Lift         | Can          | Square       | ALOHA Cube   | Average |
> |----------------------------|--------------|--------------|--------------|--------------|---------|
> | DP                         | 0.60 +- 0.00 | 0.63 +- 0.01 | 0.48 +- 0.00 | 0.32 +- 0.00 | 0.51    |
> | RC-DP                      | 0.40 +- 0.04 | 0.73 +- 0.03 | 0.66 +- 0.02 | 0.60 +- 0.04 | 0.60    |
> | DP+Repr                    | 0.66 +- 0.04 | 0.61 +- 0.01 | 0.44 +- 0.02 | 0.25 +- 0.03 | 0.49    |
> | DP PT + FT                 | 0.52 +- 0.02 | 0.67 +- 0.01 | 0.57 +- 0.03 | 0.78 +- 0.00 | 0.64    |
> | UniPi-OL                   | 0.12 +- 0.06 | 0.28 +- 0.02 | 0.07 +- 0.01 | 0.00 +- 0.00 | 0.12    |
> | UniPi-CL                   | 0.12 +- 0.02 | 0.30 +- 0.02 | 0.10 +- 0.04 | 0.15 +- 0.07 | 0.17    |
> | LDP                        | 0.69 +- 0.03 | 0.70 +- 0.02 | 0.46 +- 0.00 | 0.64 +- 0.04 | 0.65    |
> | LDP + Subopt               | 0.84 +- 0.06 | 0.68 +- 0.02 | 0.55 +- 0.03 | 0.71 +- 0.03 | 0.70    |
> | LDP + Action-Free + Subopt | 1.00 +- 0.00 | 0.98 +- 0.00 | 0.83 +- 0.01 | 0.97 +- 0.01 | 0.95    |
>
> **Q3: Training the inverse dynamics on suboptimal trajectories...**
>
> We believe this does help, because LDP + Subopt > LDP, and LDP + Action-Free + Subopt > LDP + Action-Free. In both of these cases, the difference is the suboptimal data for the IDM, which improves policy rollouts. One metric we can use to quantitatively measure this is action MSE for a validation dataset, but we found that this is noisy and not extremely correlated with the policy performance, which is true for many behavior cloning methods. Thus, we don’t find action MSE metrics for the IDM to be particularly insightful, and instead refer to policy performance.
>
> **Q4: Are the action-free data always from expert policies? To my understanding, suboptimal action-free trajectories cannot be used because they would decrease the performance of the planner.**
>
> Yes, action-free data assumes that optimal behavior.

---

### Official Review · Reviewer_zcGe · 2025-03-11

**Overall Recommendation:** 2

**Summary:**

- This paper ultimately aims to do some form of imitation learning in robotic settings
- It does this with a modular approach, using: 1) a 'planner' to predict sequences of observations from those provided by an expert demonstrator. 2) an IDM predicting actions from past and future observations.
- At inference, the planner is used to generate a sequence, which the IDM converts to actions, which can be executed.
- Both models operate on observations that are compressed by a beta-VAE. Both are trained with diffusion as the loss.
- The paper emphasizes the types of data that can be used to train each model. The planner can also be trained on non-labelled (but still expert) data, the IDM can be trained sub-optimal (but still labelled) data.
- Experiments in several simulated robotics tasks and one real robotic task show slight improvements over baselines.

**Claims And Evidence:**

See strengths/weakenesses.

**Essential References Not Discussed:**

Diffusion policy is one example of diffusion for imitation learning, but there are other relevant papers missed.

- Diffusion policies as an expressive policy class for offline reinforcement learning
- Imitating Human Behaviour with Diffusion Models
- ...

Also,
- DP-VPT is presented as a baseline, but the Tobari paper might be cited here (it is elsewhere) and used in the name, since it came well before the VPT paper.

**Experimental Designs Or Analyses:**

- Number of trajectories was quite small in all cases (100's of trajectories), which limits the impact of the work.
- Error bars overlapping for real results in Table 3

**Methods And Evaluation Criteria:**

Fine.

**Other Comments Or Suggestions:**

NA

**Other Strengths And Weaknesses:**

Strengths
- Real robotic experiments.
- Main idea is sensible.
- Minor improvements in experiments.
- Nice justification about different types of data for different models.

Weaknesses
- My main criticism of the paper is wrt the novelty. The method feels like a repeat of unipi (and probably other works) with the modification that things are done in a vae's latent space. This just doesn't feel innovative in itself to justify acceptance. Given this is a key difference, I'd expect to see a deep investigation of how to shape this space, and analysis of speedups etc. But this is lacking.
- Improvement in experiments is minor compared to baselines (around 10%). Given the variability in results that can be caused by implementation details of baselines, I'm hesitant to believe that latent diffusion planning is really delivering some substantial gain here.
- The data size is quite small for all experiments. It'd be better to see if these techniques hold in larger (1000's or 10,000's trajectories)
- While doing things in a custom latent space might be better from a speed perspective, it means pre-trained video generation models cannot be used, which I would expect might offer a good initialization for learning from the small datasets used in this work.
- In general the implementations for various components are a little outdated -- beta-vae and ddpm.

**Questions For Authors:**

See weaknesses.

**Relation To Broader Scientific Literature:**

See strengths/weakenesses.

**Theoretical Claims:**

NA

---

> ### Author Rebuttal · Authors · 2025-04-01
>
> Dear Reviewer,
>
> Thank you for your detailed feedback on our project. We provide the requested experiments and address the comments and questions in detail below. Please let us know whether there are any other concerns you have that prevent you from increasing your score.
>
> **Q1: Improvement in Experiments**
>
> We agree the experimental results in our submitted draft show that there is around a 10% improvement for LDP. However, we have now run updated experiments where LDP leverages both action-free and suboptimal data. While BC-like methods work well in settings with a lot of optimal demonstration, LDP effectively leverages additional data sources, which we believe is a fundamental advantage. The results strongly outperform baselines now, by around 30% or more. **See Reviewer LWMgQ2**
>
> **Q2: Novelty**
>
> The novelty of this paper is to propose a simple and scalable algorithm to address learning from heterogeneous data sources like suboptimal and action-free data. Our main contribution is in showing good performance with a model-based method, which will enable future work to build on it. Further, we provide a comprehensive evaluation of alternative methods for learning from heterogeneous data, and establish a novel finding that our planning-based is particularly suited for this setting.
>
> In terms of comparison to UniPi, we agree that the method is fairly similar. However, we find that LDP strongly outperforms UniPi due to its method, with the main difference being the much lower-dimensional latent space. Due to this change in method, there is a large improvement in performance. We attach updated results below and highlighted in red in our PDF.
>
> In terms of how to shape the space, we add additional experiments using pretrained DINOv2 embeddings, which investigates how pretrained embeddings compare to LDP’s latent space. In these experiments, we directly swap out VAE embeddings with frozen DINOv2 embeddings.
>
> However, we found directly planning over the DINOv2 embeddings does not lead to good learned behaviors (0% success rate even for easy tasks), which we hypothesize is due to the challenges of planning over a large latent space (384 dimensional). Thus, as an alternative, we fix a random projection matrix that reduces the 384 dimensional feature space to 16 dimensions, matching LDP’s latent space. We find that LDP strongly outperforms using pretrained latent embeddings. **See Reviewer PyaC Q4**
>
>
> **Q3: Data Size in Experiments**
>
> We agree that using 1,000 or 10,000 trajectories would be interesting. However, most robotics benchmarks and tasks typically use 50-300 demonstrations (Diffusion Policy, Robomimic, Action Chunking with Transformers, etc.). For example. Robomimic, a popular simulated imitation learning benchmark, includes datasets with tasks that have 200 demonstrations. Simulated ALOHA tasks use only 50 demonstrations. It is typically difficult to find single-task imitation learning datasets with more than a few hundred trajectories, due to the difficulty of collecting large-scale expert demonstrations.
>
> Larger datasets are often multi-task or language-conditioned, and they are often used to train large multi-task or language-conditioned policies. (Open X-Embodiment, DROID, Octo, OpenVLA). These datasets are expensive to train on, especially for video models. We do not explore these datasets, due to the lack of good pretrained models, and because video models are much more expensive to train. This is why we are starting with small datasets, but this is an important direction for future work.
>
>
> **Q4: Pretrained Video Models**
>
> We agree that LDP cannot use pretrained video models. However, for our UniPi baseline, we compare finetuning a pretrained model vs. training from scratch. To provide a comparison, we use a model with weights pretrained on THOR, and from scratch. In these results, we find that initializing from a pretrained model does not actually improve performance. **See Reviewer 9mX6 Q1**
>
> The power of pretrained video models may lie in its ability to generalize and extrapolate to new tasks and scenes, often through language conditioning, which we leave to future work. Generalization is important for robot learning, but we find that for single-task imitation learning, which typically use smaller datasets of 50-300 demonstrations, using pretrained video models is not essential.
>
> **Q5: Outdated Beta-VAE and DDPM**
>
> We agree that there are advances to beta-VAE (VQ-VAE, VQ-GAN, etc.) and DDPM (DDIM, Consistency Models, etc.). However, we choose both of these implementations due to their simplicity. For our planner, fast inference is not crucial to our contribution, and hence, we don’t use faster samplers like DDIM. However, we agree that improvements to both beta-VAE and DDPM can lead to additional improvements and scalability, which we leave to future work.
>
> **Q6: Citations**
>
> Thank you for those references. We will include them in the draft.

---

> > ### Comment · Reviewer_zcGe · 2025-04-03
> >
> > Thank you for your response -- I appreciate a lot of hard work went into the rebuttal. I am inclined to maintain my score for now, but will engage with other reviewers in an open-minded manner.

---

> > > ### Author Response · Authors · 2025-04-08
> > >
> > > Thank you for taking our further results and ablations into consideration. We appreciate your feedback, and we plan on incorporating writing suggestions from reviewers and our updated results in our paper.
> > >
> > > To address one of your comments again:
> > >
> > > **Q3: Data size**
> > >
> > > An additional way we can test our method is on the LIBERO dataset [1]. This is a multi-task simulated dataset with 130 tasks with 50 human-collected demonstrations each. One way to evaluate LDP's performance on larger robotic datasets is pretraining on many tasks and finetuning on a single downstream task. We can also use data from other tasks as suboptimal data for the other task, in order to learn representations or dynamics. Given that we have reached the end of the rebuttal period, unfortunately we cannot include results, but we may look forward to including this in a camera-ready version. Please let us know if you may find this compelling.
> > >
> > > [1] Liu, Bo, et al. "Libero: Benchmarking knowledge transfer for lifelong robot learning." Advances in Neural Information Processing Systems 36 (2023): 44776-44791.
> > >
> > > Thank you again for your time in providing feedback for the paper!

---

### Official Review · Reviewer_PyaC · 2025-03-12

**Overall Recommendation:** 4

**Summary:**

The work proposes a novel approach for imitation learning that combines an inverse dynamics model (IDM) with a planner that proposes future goal states in latent space. The approach first trains a variational autoencoder (VAE) that encodes visual representations of states into a lower dimensional latent space. Using the embeddings from this encoder, a IDM model is trained that given a state representation and future state representation predicts the action that will lead to that future state. At the same time, a planner is trained that predicts a sequence of embeddings of future states given an embedding of a state. The main insight of this work is that the IDM model can be trained with suboptimal/ general data that contains states and actions within the given environment, and the planner can be trained with (ideally optimal) demonstrations of solving the desired task but no actions are required for training the planner. Both the IDM and planner models are implemented using diffusion models. The approach is evaluated in a series of simulated robotics tasks and a real world robotics experiment. Compared to a series of imitation learning and planning baselines and ablations, the proposed approach is found to be more effective at leveraging data that is partly suboptimal and/ or action-free.

**Claims And Evidence:**

The claims made in this work are largely clearly presented and well supported through empirical evidence. However, there are several inconsistencies in the experimental evaluation that I would expect to be corrected or well justified since these might invalidate some of the findings of the provided experiments (see Experimental Design and Analyses, esp. 1. and 3.)

**Essential References Not Discussed:**

I am not aware of any essential references that require further discussion.

**Experimental Designs Or Analyses:**

The work uses vastly different amounts of demonstrations across all tasks in the evaluation
- Can, Square: 100 demonstrations, 500 suboptimal trajectories, 100 action-free trajectories
- Lift: 3 demonstrations, 500 suboptimal trajectories, 100 action-free trajectories
- Transfer Cube: 25 demonstrations, 500 suboptimal trajectories, 25 action-free trajectories
and the amount of suboptimal trajectories in particular is significantly larger than the main demonstrations provided in the Lift and Transfer Cube tasks. In the ALOHA Transfer Cube task, it is clear that leveraging the 500 suboptimal trajectories is necessary for good performance from the results in Table 2. In particular typical DP is unable to perform well in this task since it is only trained on 25 demonstrations but using suboptimal trajectories via reward conditioning (RC-DP) or pre-training (DP PT + FT) significantly improves upon DP and in particular in the latter case performs comparable to LDP.

1. Would the authors be able to explain why these vastly different amounts of trajectories were chosen?
2. The work provides several ablations on varying data being used across its experiments and compares to a large set of sensible baselines, but it does not ablate the VAE component in its experiments. I would expect that directly planning in high-dimensional image-space performs worse than the proposed approach in latent space, but the work provides no direct evidence for this claim.
3. From the supplementary material, I see several differences between LDP and baseline algorithms or ablations. Would the authors be able to explain the following differences and provide fair comparisons in these tasks?
	1. According to Table 5, LDP trains a larger network for the ALOHA cube task compared to other baselines (esp. DP and DP-based algorithms). Why do you use larger networks for LDP and could you provide comparisons to DP at the same size of policy network to ensure that LDP does not outperform the baselines in this task due to its larger networks?
	2. According to Table 6, LDP trains a larger IDM model than the UniPi baselines in the Can task (5 vs 3 blocks) and trains its IDM model for longer in all tasks compared to UniPi (500k vs 200k gradient steps). Again, what is the reason for these discrepancies? For fair comparisons, I would expect models to be trained for similar amounts of steps and models to be of identical size where possible.
	3. According to Appendix section A.2, LDP + Subopt is trained with 50% optimal and 50% suboptimal data. Would the authors be able to clarify what they mean? Is each update batch constructed of optimal and suboptimal data in equal proportions, or do you subsample the dataset of optimal and suboptimal demonstrations to have an equal share of both?
	4. According to Appendix section A.2, LDP + Action-Free trains the main IDM model only on a single expert demonstration. Why do you not train the IDM model still on all available demonstrations as done for other algorithms?
	5. According to Appendix section A.2, LDP Hierarchical uses a smaller IDM model compared to LDP. This renders the comparison in Table 4 as unfair and not convincing anymore to state that dense forecasting is an important contribution and part of LDP.

**Methods And Evaluation Criteria:**

The work strongly relies on the latent space learned by the VAE to be expressive and representative of features that would be important for learning the dynamics and decision making within the task. Given the efficacy of pre-trained visual encoders for imitation learning (e.g. [1, 2, 3, 4]), I wonder whether this step of training a task-specific VAE is even necessary or if you could replace this with an off-the-shelf pre-trained encoder (and if necessary fine-tune on available data). Establishing that this work could work well with pre-trained visual encoders would further generalise the applicability of this approach and reduce training cost.

[1] Nair, Suraj, Aravind Rajeswaran, Vikash Kumar, Chelsea Finn, and Abhinav Gupta. "R3m: A universal visual representation for robot manipulation." arXiv preprint arXiv:2203.12601 (2022).

[2] Schäfer, Lukas, Logan Jones, Anssi Kanervisto, Yuhan Cao, Tabish Rashid, Raluca Georgescu, Dave Bignell, Siddhartha Sen, Andrea Treviño Gavito, and Sam Devlin. "Visual encoders for data-efficient imitation learning in modern video games." arXiv preprint arXiv:2312.02312 (2023).

[3] Shang, Jinghuan, Karl Schmeckpeper, Brandon B. May, Maria Vittoria Minniti, Tarik Kelestemur, David Watkins, and Laura Herlant. "Theia: Distilling diverse vision foundation models for robot learning." arXiv preprint arXiv:2407.20179 (2024).

[4] Yuan, Zhecheng, Zhengrong Xue, Bo Yuan, Xueqian Wang, Yi Wu, Yang Gao, and Huazhe Xu. "Pre-trained image encoder for generalizable visual reinforcement learning." Advances in Neural Information Processing Systems 35 (2022): 13022-13037.

**Other Comments Or Suggestions:**

No other comments or suggestions.

**Other Strengths And Weaknesses:**

I would like to commend the authors on a overall well constructed and clearly motivated evaluation. In particular, all the baselines serve specific purposes and allow to identify the importance of different components of LDP.

**Questions For Authors:**

1. The supplementary material (A.2, Table 5 and 6) reveal several inconsistencies across LDP and its baselines or ablations in several experiments. I would expect the author to correct these inconsistencies and/ or provide convincing justifications for them. Otherwise, it is unclear whether the empirical findings are due to the proposed algorithms or differences in hyperparameters. (see 3. in Experimental Design and Analyses for more details)
   I otherwise consider this a strong paper and will increase my score if the authors are able to provide convincing justification or corrections for these inconsistencies.
2. Would the authors be able to explain why the varying amounts of trajectories used to train LDP and baseline algorithms across tasks? (see Experimental Design and Analyses for more details)
3. Would the authors be able to provide an ablation that does LDP with its planner and IDM model directly operating on images rather than the VAE latent space? I would expect such an ablation to perform worse but the work currently provides no evidence for such claims.

**The score has been updated in response to the author rebuttal**

**Relation To Broader Scientific Literature:**

The work does a good job at relating to alternative diffusion-based and IDM-based imitation learning algorithms. In particular, it refers to important literature that leverages action-free or suboptimal data but clarifies the distinctions to LDP.

**Theoretical Claims:**

There are no theoretical contributions or proofs to check.

---

> ### Author Rebuttal · Authors · 2025-04-01
>
> Dear Reviewer,
>
> Thank you for your detailed feedback on our project. We are happy that you found the paper well-presented with clear experimental evaluations. Please let us know whether there are any other concerns you have that prevent you from increasing your score.
>
> **Q1: Would the authors be able to explain why these vastly different amounts of trajectories were chosen?**
>
> For different robotics tasks, we tried to keep most of the parameters the same: 100 action-free trajectories for robomimic tasks and 500 suboptimal trajectories for all tasks.
>
> \# of demonstrations: Robomimic includes 200 proficient demonstrations per task, and ALOHA includes 50 demonstrations per task. To showcase the effectiveness of our method, we always take **half** the number of demonstrations-- 100 for Robomimic tasks, and 25 for ALOHA tasks. This is a consistent protocol we use to select the number of demonstrations.
>
> \# of action-free trajectories: we chose to use the remaining **half** of the demonstrations as action-free trajectories
> Lift is a very simple task, so we reduce the number of demonstrations to 3 increase the complexity.
>
> **Q2: Ablating the VAE Component**
>
> To address “directly planning in high-dimensional image-space,” we include the UniPi baseline, which learns a video prediction planner. UniPi-CL, in particular, mirrors LDP’s planner, in that UniPi-CL forecasts over dense states instead of subgoals. Let us know if that addresses your concern!
>
> **Q3.1 DP vs. LDP on ALOHA Cube**
>
> Specifically for ALOHA Cube, we found that a larger planner was imperative for reasonable performance. We did not find this necessary for DP, and thus, we kept the smaller DP architecture. We have updated results where DP is trained with the same architecture size as all LDP variants (down_dims = [512, 1024, 2048]). We use batch_size=16 for DP, since the end-to-end training of the encoder requires much more GPU memory than LDP, which uses frozen embeddings. We couldn't run DP with a larger batch size or down_dims.
>
> | DP           | LDP          | LDP + Action-Free | LDP + Subopt | LDP + Action-Free + Subopt |
> |--------------|--------------|-------------------|--------------|----------------------------|
> | 0.32 +- 0.00 | 0.64 +- 0.04 | 0.70 +- 0.02      | 0.71 +- 0.03 | 0.97 +- 0.01               |
>
>
> **Q3.2 LDP vs. UniPI: IDM size for Can, and IDM train time**
>
> We observed a bigger IDM is helpful for the Can task, likely because the scene is slightly more visually complex. As requested, we have now rerun Can experiments with UniPi to ensure the same IDM architecture.
>
> We train the IDM model for longer in LDP because we use a smaller batch size. However, we agree that for consistency, rerun UniPi experiments with the same batch size and \# gradient steps for LDP and UniPi IDM. In addition, we have retrained UniPi GCBC models with the same hyperparameters as LDP. **See Reviewer LWMg Q2.**
>
> **Q3.3 LDP 50% optimal and 50% suboptimal batches**
>
> Each batch consists of 50% optimal and 50% suboptimal trajectories.
>
> **Q3.4 IDM one expert demonstration**
>
> We meant to write, “The IDM is trained only **ON** expert demonstrations.” The IDM is only trained on action-labelled data, which in this case, are from the expert demonstrations.
>
> **Q3.5 LDP Hierarchical IDM**
>
> (Appendix A.2) Hierarchical LDP’s IDM is a Conditional U-Net with down-dims [256, 512], which has 1.67e7 parameters. (Appendix A.1) Non-Hierarchical LDP uses an MLP ResNet based off of IDQL, which has 1.79e6 parameters. The Hierarchical LDP IDM has more parameters, while still underperforming our method.
>
> **Q4: Pretrained Encoders**
>
> LDP uses a task-specific VAE, because existing image encoders or VAEs (e.g. DINOv2, R3M, Stable Diffusion VAE) produce high-dimensional embeddings. Our VAE produces a much more compact latent space (16 dimension), enabling much faster training and inference, as well as interpretability through decoding image latents.
>
> To address this approach, we swap out VAE embeddings with frozen DINOv2 embeddings.
>
> However, we found directly planning over the DINOv2 embeddings does not lead to good learned behaviors (0% success), which we hypothesize is due to the challenges of planning over a large latent space (384 dim). Thus, as an alternative, we fix a random projection matrix, reducing the 384 dim feature space to 16 dim, matching LDP. We choose this, because this is a straightforward way of using frozen embeddings. It may be possible to plan over the large embeddings space with a much larger and complex model, or it may be possible to learn alternative ways to project DINOv2 embeddings to a lower-dimensional space, but this may lead to fundamental changes in our method, so we don’t explore more complicated approaches to use pretrained embeddings.
>
> | Method | Lift         | Can          | Square       |
> |--------|--------------|--------------|--------------|
> | DINOv2 | 0.44 +- 0.24 | 0.03 +- 0.01 | 0.01 +- 0.01 |

---

> > ### Comment · Reviewer_PyaC · 2025-04-07
> >
> > I thank the authors for their response and clarifications. They help greatly in increasing my confidence in the submission. In particular, my primary concerns about inconsistencies in the evaluation have been addressed and therefore I have decided to increase my score to **accept**.
> >
> > The evaluation protocol and clarifications about baselines / ablations are very helpful and I hope the authors will be able to incorporate these into their work (or at least the appendix). I also believe that their investigation to use DINOv2 embeddings is very interesting and adds further insights so I would suggest to include it in the appendix with a small note in the main paper.
> >
> > One outstanding comment re Q2: I believe there are sufficient differences between LDP and the UniPi baseline to justify adding such a separate ablation of the image encoder VAE. I agree with your intuition that the results of UniPi (and prior work) suggests that directly learning in image space is expected to fare much worse and, thus, I don't see this as essential. Nevertheless, it would be a nice comparison point to bring this point home in a convincing manner.

---

> > > ### Author Response · Authors · 2025-04-08
> > >
> > > Thank you for taking our further results and ablations into consideration and increasing your score! We plan on incorporating writing suggestions from reviewers and our updated results in our paper. We appreciate your feedback and are open to any other comments or suggestions that can help improve our draft.
> > >
> > > We agree the ablation could still provide further insight into the method. One of our main challenges is computational constraints, and hence we focused on UniPi as an image-planning baseline. We may try to include an ablation, in addition to our rebuttal experiments, in an updated draft.

---

### Official Review · Reviewer_9mX6 · 2025-03-21

**Overall Recommendation:** 3

**Summary:**

This paper presents Latent Diffusion Planning (LDP), an algorithm aimed at performing imitation learning with the presence of additional suboptimal and action-free demonstrations.

----

Problem Setting and Key Assumptions:
- Vision-based imitation learning for table top manipulation
- Aside from expert demos, assume access to (expert) action-free demos and suboptimal/failed data

----

The main algorithm consists of three components:
- A VAE to learn a low-dimensional embedding space for the visual observations
- A diffusion-based planner that performs forward prediction in the frozen VAE embedding space
- A diffusion-based inverse dynamics model that takes adjacent VAE latents to predict robot actions.
This design allows the method to learn from the three types of data as mentioned earlier. Specifically, the VAE is trained to enc/dec individual frames, and thus, can take all the data. The planner only need the optimal sequence of latent states. Hence, both expert and action-free demos can be used. Finally, the inverse dynamics can be trained on all $(o, a, o')$ tuples, regardless of whether they lead to task success or not.

----

The authors evaluate LDP in 4 simulated tasks and a real robot task on tabletop manipulation. Because the assumption on the available data is new, they compare with Diffusion Policy and a few variants. They also compare with UniPi. Results support that in the low-data setting, LDP outperforms these relevant baselines by also utilizing the action-free demo and suboptimal data.

----

## Update after Rebuttal

I reviewed other reviewers' comments and the authors' responses. I have raised my score from 2 to 3.

**Claims And Evidence:**

The main claim is that LDP should be data efficient (for expert data) by leveraging sub-optimal data and action-free data. This is indeed verified with experiments. The main weakness is that the algorithm still seems to require many demonstrations (and more suboptimal data) for rather simple tasks.

**Essential References Not Discussed:**

N/A

**Experimental Designs Or Analyses:**

The experiment procedure and selected baseline methods on imitation learning are fair. Regarding the real robot experiment, the authors provide action-free demonstration by removing actions from actual demonstrations. This is a rather awkward design choice. Instead, they should consider collecting demonstrations using a setup similar to the Universal Manipulation Interface paper. In that case, I wonder if the planner model can still reliably predict because the data distribution of action-free demo will be different from the expert demos.

**Methods And Evaluation Criteria:**

Overall, the method and evaluation procedures are sound. However, the experiment results are not convincing enough because:
- The comparison with a foundation model like UniPi feels somewhat out of place because I assume the authors train the model from scratch using only in-domain data. Fine-tuning a pre-trained model and allowing joint training on multiple datasets are more appropriate.
- The tasks are rather simple with clean visuals.

**Other Comments Or Suggestions:**

N/A

**Other Strengths And Weaknesses:**

The paper is easy to follow and includes clear visualizations of the model architecture, and comprehensive discussion of relative work. The experiments sections presents the research questions clearly followed by the corresponding studies.

**Questions For Authors:**

For the real robot experiment, how much time does it take the collect the demonstrations? Recent works have shown that similar tasks can be learned from 1 hour of data, including a few demonstrations and sparse-reward RL. (Accelerating Visual Sparse-Reward Learning with Latent Nearest-Demonstration-Guided Explorations​)

**Relation To Broader Scientific Literature:**

This work aims to improve the data efficiency of IL methods by utilizing data otherwise not useful. It is highly relevant to the field of LfD (Learning from Demonstrations).

**Theoretical Claims:**

No theoretical claims.

---

> ### Author Rebuttal · Authors · 2025-04-01
>
> Dear Reviewer,
>
> Thank you for your detailed feedback on our project. We are happy that you found the paper easy to follow and relevant to learning from demonstrations. Please let us know whether there are any other concerns you have that prevent you from increasing your score.
>
> **Q1: Pretrained UniPi + Finetuning In-Domain**
>
> The comparison with UniPi is limited as the original UniPi model, code and pretraining data are not released, per Appendix A.1. To provide a comparison, we use AVDC-THOR, which is the best available reproduction of UniPi. We include results on Robomimic tasks. The experiments compare how finetuning a pretrained UniPi model, instead of training a UniPi model from scratch, affects overall performance. We finetune for an additional 50k steps at learning rate 1e-4.
>
> | Method                   | Lift         | Can          | Square       |
> |--------------------------|--------------|--------------|--------------|
> | UniPi-OL (from scratch)  | 0.09 +- 0.03 | 0.27 +- 0.01 | 0.07 +- 0.01 |
> | UniPi-OL (from pretrain) | 0.13 +- 0.05 | 0.31 +- 0.01 | 0.07 +- 0.01 |
> | UniPi-CL (from scratch)  | 0.12 +- 0.02 | 0.30 +- 0.02 | 0.10 +- 0.04 |
> | UniPi-CL (from pretrain) | 0.13 +- 0.05 | 0.23 +- 0.01 | 0.08 +- 0.02 |
>
> | Method                                 | Lift         | Can          | Square       |
> |----------------------------------------|--------------|--------------|--------------|
> | UniPi-OL + Action-Free (from scratch)  | 0.11 +- 0.03 | 0.25 +- 0.03 | 0.05 +- 0.03 |
> | UniPi-OL + Action-Free (from pretrain) | 0.10 +- 0.08 | 0.26 +- 0.00 | 0.08 +- 0.02 |
> | UniPi-CL + Action-Free (from scratch)  | 0.14 +- 0.02 | 0.32 +- 0.04 | 0.09 +- 0.01 |
> | UniPi-CL + Action-Free (from pretrain) | 0.11 +- 0.03 | 0.25 +- 0.01 | 0.10 +- 0.00 |
>
> We observe the pretrained AVDC/UniPi model does not substantially improve performance, likely because the pretrained model quality is limited. For UniPi-OL, there appears to be slightly improvement when pretraining from scratch, possibly because without closed-loop planning, the pretrained initialization may be helpful.
>
> **Q2: Action-Free Demonstrations**
>
> Thank you for this suggestion. We were unable to set up the UMI hardware within the short rebuttal time, but we agree that a more scalable and realistic way of obtaining action-free expert data is through data collection tools such as UMI (Universal Manipulation Interface).
>
> **Q3: Collecting Demonstrations in the Real World**
>
> The demonstrations were collected from one teleoperator. The average time for one demonstration was around 25 seconds, including resetting the cube. In total, for 82 trajectories, it would take around 34 minutes to collect all demos.
>
> For suboptimal trajectories, the teleoperator merely supervises the policy to ensure safety, and does not need to actually teleoperate the robot at all. Each suboptimal trajectory takes around 75 seconds, including resets. For 85 trajectories, this would take around 1 hour 45 minutes.
>
> The total time is longer than LANE from Zhao et. al., which collects limited demonstrations (requires teleoperator) and performs RL (does not require active teleoperation). It’s possible that our method may work with fewer demonstrations (less teleoperation time) or with less suboptimal data, but we did not sweep over these numbers or optimize for these metrics.
>
> **Q4: Additional Comments**
>
> “The tasks are rather simple with clean visuals.” -- We agree that the tasks are not extremely complex nor do they have complex backgrounds or visuals. However, we chose commonly used robotics benchmarks for this project and focused on showing the effect of action-free and suboptimal data.
>
> “The main weakness is that the algorithm still seems to require many demonstrations (and more suboptimal data) for rather simple tasks.” -- We agree that this paper doesn’t operate in a few-shot demonstration setting. However, the standard number of demonstrations provided by robomimic is 200 per task, and ALOHA is 50 per task. The robomimic benchmark is saturated with 200 demos, and we chose to restrict the number of demonstrations to evaluate our method in a lower demonstration regime. In order to learn from even fewer demonstrations for these tasks, it would likely be necessary to learn from prior data (e.g. from retrieval), or reinforcement learning, which we do not focus on.
>
> **Q5: Improved Experiments**
>
> Please **see Reviewer LWMg Q2**. We include updated experiments where LDP leverages both action-free and suboptimal data. While BC-like methods work well in settings with a lot of optimal demonstration, LDP effectively leverages additional data sources, which we believe is a fundamental advantage. The results strongly outperform baselines now, by around 30% or more.
>
> In addition, we have reran UniPi with identical hyperparameters to LDP, per Reviewer PyaC’s suggestion.

---

> > ### Comment · Reviewer_9mX6 · 2025-04-07
> >
> > I appreciate the authors for performing additional studies around UniPi. The new experiments results in response to reviewer LWMg are also convincing.
> >
> > I'm willing to increase my score to 3.

---

> > > ### Author Response · Authors · 2025-04-08
> > >
> > > Thank you for taking our further results and ablations into consideration and increasing your score! We appreciate the feedback and are open to any other comments or suggestions that can help improve our draft.

---

### Decision · Program_Chairs · 2025-05-01

**Decision:**

Accept (spotlight poster)

**Comment:**

The paper introduces Latent Diffusion Planning (LDP), a modular imitation learning framework that separates planning and control by operating in a learned latent space. It combines a planner trained with action-free demonstrations and an inverse dynamics model trained with suboptimal, action-labeled data. Both modules are trained using diffusion-based objectives and rely on a variational autoencoder (VAE) to encode image-based observations into compact latent representations. The key insight is that decoupling planning from action prediction allows each module to exploit broader, weaker forms of supervision. Experimental results on simulated and real-world robotic manipulation tasks demonstrate improved performance and data efficiency compared to existing imitation learning methods, especially in settings with limited expert demonstrations.

The paper offers a creative modular solution for leveraging imperfect demonstration data. The use of a latent space to enable diffusion-based planning and control is intuitive and allows the effective reuse of suboptimal and action-free demonstrations. Empirical results are generally favorable, and real-world experiments strengthen the practical relevance. However, the novelty is somewhat incremental, given similarities to prior works like UniPi, and the improvements over baselines are modest. Experimental inconsistencies—such as varying model sizes, training steps, and dataset sizes across methods—raise concerns about the fairness of some comparisons. Additionally, the lack of ablation for the VAE and limited use of larger-scale datasets or modern pre-trained visual encoders limits the generality and robustness of the findings. However, most of the concerns were addressed by the author's rebuttal, leading to the increased scores.

In sum, I recommend accepting this paper.